# Underwater High-Precision 3D Reconstruction System Based on Rotating Scanning

**DOI:** 10.3390/s21041402

**Published:** 2021-02-17

**Authors:** Qingsheng Xue, Qian Sun, Fupeng Wang, Haoxuan Bai, Bai Yang, Qian Li

**Affiliations:** College of Information Science and Engineering, Ocean University of China, Qingdao 266100, Shandong, China; xueqingsheng@ouc.edu.cn (Q.X.); 21190231177@stu.ouc.edu.cn (Q.S.); baihaoxuan@stu.edu.cn (H.B.); liqian@ouc.edu.cn (Q.L.)

**Keywords:** 3D reconstruction, underwater laser scanning, laser sensor, high precision

## Abstract

This paper presents an underwater high-precision line laser three-dimensional (3D) scanning (LLS) system with rotary scanning mode, which is composed of a low illumination underwater camera and a green line laser projector. The underwater 3D data acquisition can be realized in the range of field of view of 50° (vertical) × 360° (horizontal). We compensate the refraction of the 3D reconstruction system to reduce the angle error caused by the refraction of light on different media surfaces and reduce the impact of refraction on the image quality. In order to verify the reconstruction effect of the 3D reconstruction system and the effectiveness of the refraction compensation algorithm, we conducted error experiments on a standard sphere. The results show that the system’s underwater reconstruction error is less than 0.6 mm within the working distance of 140 mm~2500 mm, which meets the design requirements. It can provide reference for the development of low-cost underwater 3D laser scanning system.

## 1. Introduction

In recent years, three-dimensional (3D) terrain data and scene reconstruction technology have been gradually applied to underwater imaging applications. Due to the disadvantages of traditional acoustic detection, such as it being easily affected by underwater noise, non-intuitive imaging, and poor visualization effect, optical 3D reconstruction technology gradually plays its advantages in underwater operations. For example, in terms of underwater engineering construction [1], laser 3D reconstruction technology can provide more accurate and visualized 3D site surveys for underwater construction projects, used to check the structure of subsea instruments and the wear status of pipelines. In military applications [2], laser 3D reconstruction technology can be used to detect and monitor the underwater targets, providing real-time data support for underwater military rescue. In terms of marine scientific research, laser 3D reconstruction technology can explore seabed resources and map seabed topography [3,4]. In addition, it can also be applied to biological survey [5,6], archaeology [7,8,9,10], sea bottom topography description, etc. [11,12].

Linear laser 3D reconstruction is an active optical measurement technology that belongs to structured light 3D reconstruction technology. The basic principle is that the structured light projector projects a controllable light strip onto the surface of the object to be measured, the image is obtained by the image sensor (such as camera), and the 3D coordinates of the object are calculated by the triangulation through the geometric relationship of the system [13]. Also, there are light spot and light beam laser scanning technology [14]. The light spot type is to scan the object point by point; however, the disadvantage is that image acquisition and post-processing are time-consuming and difficult to complete real-time measurement. Light plane type is to project a two-dimensional plane to an object, and the measurement speed is the fastest. It is usually used to project grating stripes, which is usually used in the air. The linear laser scanning sensor system is composed of a low-light underwater camera, a green line laser projector, and a scanning turntable. The calibration of the system includes the calibration of camera parameters and the calibration of system structure parameters. Through calibration, we can obtain the conversion relationship between the two-dimensional pixel coordinate system and the three-dimensional camera coordinate system, and then obtain the equation of the laser stripe on the plane target in the three-dimensional space. A series of solving and fitting algorithms can finally obtain the plane equation of the light plane in the camera coordinate system. In the early stage of linear laser sensor calibration, the mechanical method was proposed first and widely used. Later, the idea based on the cross ratio invariant method was developed to get rid of the shackles of precision platform in principle. A certain number of high precision calibration points can be obtained by using the specially designed stereo target to complete the calibration [15,16,17,18]. Other methods, such as hidden point, three-point perspective model, active vision, and binocular stereo vision, have certain applications on some specific occasions, greatly enriching the calibration method of the linear structured light sensor [19,20,21,22,23,24].

Michael Bleier et al. developed a laser scanning system with the wavelength of 525 nm to scan the objects in the water tank statically and dynamically. From the obtained point cloud data, the reconstruction errors of the two state systems are less than 5 cm [25]. Jules S. Jaffe [26] invented a line laser scanning imaging system for underwater robots, which effectively reduces backscatter and volume scattering by using large-scale camera splitting, scanning, or pulse systems. Yang Yu et al. [27] studied a multi-channel RGB laser scanning system and proposed a high-resolution underwater true-color three-dimensional reconstruction method with three primary color lasers as the active light source, which can target objects including human faces. Scanning and reconstruction of the millimeter level are performed while restoring the true color texture information of the target. Xu Wangbo et al. [28] designed and implemented an underwater object measurement system based on multi-line structured light. When the measurement distance is 2.45 m, the average error reaches 1.6443 mm, which has high measurement accuracy. At present, many research institutions have done a lot of work on image, point cloud processing, and water scattering correction, and good experimental results have been obtained. However, in the process of underwater experiments, the influence of refraction caused by different media surfaces on the viewing angle error and imaging quality still needs more attention [29].

In order to realize the high-precision 3D reconstruction of underwater target and better utilize the advantages of the optical 3D reconstruction system in underwater operations, this paper designs and develops an underwater high-precision 3D reconstruction system based on rotary scanning, and proposes a refraction compensation method for the system. This article will describe in detail the calibration principle and system structure involved in the reconstruction process in the second and third sections. The specific experimental process will be introduced in Section 4. Then, we will perform error analysis on the standard sphere with known radius, compare the reconstruction errors of the system in two different environments of air and water, and calculate the reconstruction accuracy of underwater objects before and after using the algorithm. The specific work contents are as follows.

## 2. Determination of System Parameters and Establishment of Light Plane

In the reconstruction of laser three-dimensional scanning, the conversion of the coordinate system and a series of calibration algorithms are involved, including the calibration of the internal and external parameters of the camera, and the determination of the laser plane equation of the system. Through calibration, the two-dimensional pixel coordinates obtained by the camera can be converted into three-dimensional point cloud coordinates. In the process of obtaining the point cloud, we added the refraction correction algorithm of the waterproof device to improve the system reconstruction accuracy.

### 2.1. Coordinate System Conversion

The calibration process of the system involves the conversion between image coordinate system (including pixel coordinate system and physical coordinate system), camera coordinate system, and world coordinate system. As shown in Figure 1, plane ABC is the laser plane, and f is the camera focal length. We set $O_{W}-X_{W}Y_{W}Z_{W}$ as the world coordinate system, $O_{C}-X_{C}Y_{C}Z_{C}$ as the camera coordinate system, $O_{u}-uv$ as the pixel coordinate system, and O−xy as the physical coordinate system.

The relationship between a point $P_{1}$ in the world coordinate system and a point $P_{2}$ in the camera coordinate system is as follows:(1)$$\left[\begin{array}{c} X_{C}\\ Y_{C}\\ Z_{C} \end{array}\right]=R\left[\begin{array}{c} X_{W}\\ Y_{W}\\ Z_{W} \end{array}\right]+T,$$
where R and T are the rotation matrix and translation matrix of the camera coordinate system relative to the world coordinate system.

According to the triangle similarity principle, the relationship between the point $P_{2}$ in the camera coordinate system and $P_{3}$ in the image coordinate system can be obtained
(2)ZC[xy1]=[f0000f000010][XCYCZC1]

The conversion relationship between the point in the image coordinate system and the point in the camera coordinate system is shown in the following formula:(3)$$\left[\begin{array}{c} u\\ v\\ 1 \end{array}\right]=\left[\begin{array}{ccc} \frac{1}{d_{x}} & 0 & u_{0}\\ 0 & \frac{1}{d_{y}} & v_{0}\\ 0 & 0 & 1 \end{array}\right]\left[\begin{array}{c} x\\ y\\ 1 \end{array}\right]$$
where $d_{x}$ and $d_{y}$ represent the physical size of unit pixel in x-axis and y-axis, respectively. To sum up, the conversion relationship of points from the pixel coordinate system to the world coordinate system can be obtained:(4)ZC[uv1][1dx0u001dyv0001][f0000f000010][RT0→1][XWYWZW1]=[fx0u000fyv000010][RT0→1][XWYWZW1]

### 2.2. Camera Calibration

Camera calibration can be divided into traditional camera calibration method and camera self-calibration method [30,31]. The methods that are commonly used include the traditional Tsai calibration method and Zhang Zhengyou calibration method, which is between traditional and self-calibration methods. In this paper, the method we used is the Zhang Zhengyou calibration method. By taking photos of the calibration plate in different directions, two groups of parameters of the camera are directly obtained, which are the external parameters of spatial transformation and the internal parameters of the camera itself. Using the external and internal parameters, the corresponding relationship between the pixel coordinates of the image and the three-dimensional coordinates in the space can be established, that is, the three-dimensional space information can be obtained from the two-dimensional image [32,33]. The Zhang Zhengyou calibration method does not need to know the movement mode of the calibration plate, which avoids high equipment demand and complicated operation, and has higher accuracy and better robustness than the self-calibration method [34,35].

In the process of camera calibration, it is necessary to extract the corner coordinates of the checkerboard plane target. The most commonly used feature detection algorithms in the field of computer vision are Harris, SIFT, SURF, FAST, BRIEF, ORB, etc. The SIFT (Scale Invariant Feature Transform) algorithm proposed by David G. Lowe is based on the feature point extraction of the DoG (Difference of Gaussian) pyramid scale space [36]. The advantages are stable features and rotation invariance and the disadvantage is that the ability to extract feature points for smooth-edge targets is weak. The SURF (Speeded Up Robust Feature) algorithm is an improvement of the SIFT algorithm proposed by David Lowe in 1999 [37], which improves the execution efficiency of the algorithm and provides the possibility for the application of the algorithm in real-time computer vision systems. ORB (Oriented FAST and Rotated BRIEF) is a fast algorithm for feature point extraction and description [38,39]. It was written by Ethan Rublee, Vincent Rabaud, et al. in 2011 entitled “ORB: An Efficient Alternative to SIFT or SURF;” they proposed using the FAST algorithm for feature point extraction, and using the BRIEF algorithm to describe feature points. By combining the advantages of the FAST [40,41] and BRIEF algorithm [42,43] for fast feature point detection and simple description, and improving on them, it solves the complexity of SIFT calculation and the lack of rotation invariance, scale invariance, noise sensitivity, and other shortcomings.

This paper uses the classic Harris corner detection algorithm, which has the advantage of simple calculation and can easily recognize the gray level changes and translational rotation changes of flat images. Experimental verification shows that the Harris algorithm is suitable for the underwater rotating scanning system proposed in this paper, and can extract limited feature points on a planar target. In addition, corner points can also be accurately detected under noise interference, which has high stability and robustness. The basic idea is that the recognition of the human eye corner is usually completed in a local small area or small window [44]. If the small window with this feature is moved in all directions, the gray level of the area in the window changes greatly, and then it is considered that the corner is encountered in the window. The general steps are as follows:

Firstly, calculate the image matrix M, and each pixel of the image is filtered by using the horizontal and numerical difference operators to get the values of $I_{x}$ and $I_{y}$, and then the four elements in M are obtained:(5)$$M=\left[\begin{array}{cc} I_{x}^{2} & I_{x}I_{y}\\ I_{x}I_{y} & I_{y}^{2} \end{array}\right]$$

Then, the four elements of M are filtered by Gaussian smoothing to get a new M The discrete two-dimensional mean Gaussian function is:(6)$$Gauss=\exp\left(-\frac{\left(x^{2}+y^{2}\right)}{2\sigma^{2}}\right)$$

Use M to calculate the amount of corners corresponding to each pixel Cim:(7)$$Cim=\frac{I_{x}^{2}*I_{y}^{2}-{\left(I_{x}I_{y}\right)}^{2}}{I_{x}^{2}+I_{y}^{2}}$$

When a point satisfies that Cim is greater than the threshold and is a local maximum in a neighborhood, it can be considered as a corner.

Some of the extraction results are shown in Figure 2, and the pixel coordinates of some corner points of the calibration board are shown in Table 1.

### 2.3. Calibration of System Structure Parameters

The calibration of sensor structure parameters is the position equation of the laser plane relative to camera. The main calibration methods include the mechanical adjustment method [45], filament scattering method [46,47], and cross ratio invariant method [48]. From the angle of target, it can be divided into the three-dimensional target, plane target, one-dimensional target, and no target [49,50,51,52]. In this paper, a new method of line laser calibration is used. By combining the linear equation formed by the optical center and the point on the light strip and the plane equation of the target in the camera coordinate system, the equation of the line in the camera coordinate system can be obtained. Finally, an equation that describes the plane in the camera coordinate system can be obtained by using the least square method. Let the plane equation be Ax+By+Cz=0. The specific algorithm flow is shown in Figure 3.

### 2.4. Refraction Compensation of Underwater 3D Reconstruction System

When working underwater, the instruments are sealed in a waterproof device, so the underwater target and the laser scanning system are separated in media with different refractive indices. When the camera photographs the underwater object, the light passes through the interface of the water-plane glass waterproof cover and the glass waterproof cover-air. After being refracted twice, the object is finally being imaged on the image surface of the camera. According to the points on the image plane of the camera, we perform reverse calculations on the refraction light path to find the intersection point. This intersection point is the actual image point of the measured object after removing the effect of refraction [53,54,55,56]. Since the light is refracted twice at the waterproof cover, and the thickness of the waterproof cover is thin (negligible), the imaging process in the water is simplified as shown in Figure 4, where the plane of the waterproof cover is parallel to the imaging plane of the camera.

In view of the above situation, this section will compensate the system from two aspects. They are the offset compensation of the imaging point on the CCD object surface and the laser plane at the sealing glass.

#### 2.4.1. Refraction Compensation of Light Plane

As shown in Figure 4, the light plane A in the air is refracted by the glass surface to obtain the underwater light plane B. $\theta_{1}$ is the laser projection angle, $\theta_{2}$ is the angle between the light plane A and the normal, $\theta_{3}$ is the is the angle between the normal to A and the normal to C, the $\theta_{4}$ is the angle between the light plane B and the normal C, and $\theta_{5}$ is the angle between the normal B and the normal C. 

Suppose the normal vector of the refractive surface is (0,0,1), the normal vector of the light plane A is (A,B,C), the normal vector of the light plane B is $\left(A^{\prime }{,B}^{\prime }{,C}^{\prime }\right)$, and the relative refractive index of water and air is $n^{\prime }$. From the law of refraction, we can get that:(8)$$n^{\prime }=\frac{n_{1}}{n_{3}}=\frac{\sin\theta_{1}}{\sin\theta_{3}}=\frac{\cos\theta_{3}}{\cos\theta_{5}}$$
where $n_{1}$ is the refractive index of air and $n_{3}$ is the refractive index of water.
(9)$$\frac{C}{\sqrt{A^{2}+B^{2}+C^{2}}}=n^{\prime }\frac{C}{\sqrt{A^{\prime 2}+B^{\prime 2}+C^{\prime 2}}}$$

Since the normal vector of the light plane B is a unit vector, then:(10)$$\sqrt{A^{\prime 2}+B^{\prime 2}+C^{\prime 2}}=1$$

According to (6), it can be concluded that:(11)$$C^{\prime }=\frac{C}{n^{\prime }\sqrt{A^{2}+B^{2}+C^{2}}}$$
(12)$${\left[\begin{array}{c} A\\ B\\ C \end{array}\right]}^{T}=x{\left[\begin{array}{c} A^{\prime }\\ B^{\prime }\\ C^{\prime } \end{array}\right]}^{T}+y{\left[\begin{array}{c} 0\\ 0\\ 1 \end{array}\right]}^{T}$$

According to the simultaneous Formulas (11) and (12), it can be concluded that:(13)$$\{\begin{array}{c} A^{\prime }=\sqrt{\frac{A^{2}n^{\prime 2}\left(A^{2}+B^{2}+C^{2}\right)-A^{2}C^{2}}{n^{\prime 2}(A^{2}+B^{2}+C^{2})(A^{2}+B^{2})}}\\ B^{\prime }=\frac{A^{\prime }B}{A} \end{array}$$

The coordinates of intersection point D between light plane A and light plane B and glass surface are$\left(0,H\tan\theta_{1},H\right)$, where H is the distance from camera optical center to glass surface, and the equation of light plane B is as follows:(14)$$A^{\prime }x+B^{\prime }y+C^{\prime }z+D^{\prime }=0$$

Substituting the coordinate of point d into (11), we can get the following results:(15)$$\{\begin{array}{c} B^{\prime }H\tan\theta +C^{\prime }H+D^{\prime }=0\\ BH\tan\theta +CH+D=0 \end{array}$$

After simplification, we can get the following conclusions:(16)$$D^{\prime }=-(B^{\prime }\tan\theta +C^{\prime }H)=B^{\prime }\frac{CH+D^{\prime }}{B}-C^{\prime }H$$

In conclusion, the laser plane equation after refraction can be calculated:*A*’*x* + *B*’*y* + *C*’*z* + *D*’ = 0.(17)

#### 2.4.2. CCD Image Coordinate Refraction Compensation

It can be seen from the Figure 4 that the imaging point of the underwater target $P_{w}$ on the image plane is P(u,v). If it is in the air, the direct imaging point is $P^{\prime }(u^{\prime },v^{\prime })$. There is a relationship between P and $P^{\prime }$:(18)$$\left(u^{\prime },v^{\prime }\right)=\eta \left(u,v\right)$$

For each measured point, refraction correction can be realized as long as it is obtained.
(19)$$\eta =\frac{\tan\theta_{7}}{\tan\theta_{6}}=\frac{\left(EF\tan\theta_{8}+GF\right)/O_{C}F}{IJ/O_{C}J}$$
and because:(20)$$IJ=\sqrt{u^{2}+v^{2}}$$
(21)$$O_{C}J=f$$
where f is the focal length of the camera, point $P^{\prime }\left(x_{c},y_{c},z_{c}\right)$ is known; when a condition is satisfied $Z_{W}\gg H$, the results are as follows:(22)$$\eta =\frac{\left(1-\frac{H}{Z_{C}}\right)\tan\theta_{8}+\frac{H}{Z_{C}}\tan\theta_{6}}{\sqrt{u^{2}+v^{2}}/f}\approx \frac{\tan\theta_{8}}{\sqrt{u^{2}+v^{2}}/f}$$

Substituting the results of refraction compensation into the camera calibration program and coordinate conversion formula, the compensated three-dimensional coordinates can be obtained.

## 3. Rotary Scanning System

Linear laser 3D rotation system mainly includes optical system and mechanical structure. The optical system is composed of laser, camera, and LED, as it shown in Figure 5.

The distance between the camera and the laser is 250 mm, and they are, respectively, installed in two independent watertight chambers, fixed on both sides of the support frame to maintain their relative position. The turntable is controlled by the host computer, and the system can scan at a fixed position within the range of 50° × 360°. The specific model of the system is in Figure 5. Some parameters of camera, laser and system performance are shown in Table 2:

## 4. Performance Tests

### 4.1. Three-Dimensional Reconstruction in Air

We choose a checkerboard with a side length of 25 mm as the calibration target. By changing the angle of the rotation and tilt of the target, 12 pictures of the checkerboard in different poses under the camera coordinate system can be obtained. Through Zhang Zhengyou’s calibration method, the camera’s internal parameters and calibration results include external parameters and distortion coefficients. The whole calibration process includes the original image acquisition, chessboard corner extraction, reprojection error, and other steps. The result is shown in Figure 6.

We put the fixture in the air for a scanning experiment. Through the coordinate transformation and the solution of the light plane equation, we can finally get the three-dimensional point cloud coordinates of the fixture. After noise reduction, the final three-dimensional point cloud of the fixture is shown in the Figure 7b.

### 4.2. 3D Reconstruction of Underwater Target

From the above experimental results, the system can complete the 3D point cloud coordinate extraction of the water target, and reconstruct the characteristics of the object, which verifies the feasibility of the system in the air. The next step is to put the scanning system in water and calibrate it again. It is known that light will refract on passing through different media, and underwater calibration can better reduce the influence of laser light refraction on the reconstruction accuracy, and can better scan the conch in the water to verify the reconstruction effect of the underwater system. First, we calibrate the chessboard target underwater, as shown in Figure 8.

After calibration, the internal parameter matrix of the camera is as follows:(23)$$\mathrm{Intrinsic}\;\mathrm{Matrix}=\left[\begin{array}{ccc} 12329 & 0 & 0\\ 0 & 12290 & 0\\ 1314 & 748 & 1 \end{array}\right]$$

Other related parameters include camera focal length, principal point, image size, radial distortion, and tangential distortion, as shown in the Table 3 below:

After getting the calibration results of the system structure parameters, we fit the laser plane projected by the laser. Finally, the laser plane equation is obtained as follows:(24)Z=1498.888393+0.019092x+0.1281023y

The parameters corresponding to the light plane equation Ax+By+Cz=0 in Section 2.3 are: A = 0.019092; B = 0.128103; C = 1; D = 1498.888393.

We selected the conch as the scanning object, as shown in the Figure 9a. The 3D point cloud information of a single frame image can be obtained by converting the two-dimensional image data into three-dimensional point cloud coordinates through system parameter calibration. According to the motion mode of the turntable, the three-dimensional point cloud data can be spliced. Finally, the laser scanning results are obtained. The three-dimensional point cloud image of conch is as follows:

### 4.3. Error Analysis

In order to measure the accuracy of the system, a standard ball with a radius of 20.01 mm is scanned and reconstructed in the range of 500 mm to 1200 mm. The two-dimensional coordinates of the scanned object surface are obtained by using the laser stripe center extraction algorithm, and then the three-dimensional point cloud coordinates of the standard sphere are obtained by using the coordinate transformation formula. The standard sphere radius can be obtained by calculation, and the fringe extraction result is shown in Figure 10.

The reconstruction results in air and underwater are shown in Figure 11. The average measurement error of the standard sphere in the air is less than 0.16 mm, and the average error after refraction corrected in water is less than 0.6 mm. In addition, we can also conclude that as the measurement distance increases, the error gradually increases.

In order to verify the effectiveness of the underwater refraction correction algorithm, we use different methods to carry out underwater measurement experiments, for example, we select a standard ball with a radius of 20.01 mm as the measured object, select 12 different positions in the field of view for measurement, and calculate the measurement radius of the standard ball without refraction correction algorithm and with refraction correction algorithm, respectively. R is the underwater measurement radius without the refraction correction algorithm, and $R_{c}$ is the measurement radius with the refraction correction. The detailed measurement results are shown in Table 4:

It can be seen from Table 4 that the measurement error of standard ball is within 7.1 mm without refraction correction, and the maximum measurement error is 7.099 mm. The average radius of standard sphere is 25.780 mm. After adding refraction correction, the maximum error of reconstructed standard sphere radius is 0.596 mm, and the error range is within 0.6 mm. It can be seen that the reconstruction accuracy of the system has been significantly improved after adding the refraction correction algorithm. It is proved that the algorithm proposed in this paper is accurate and effective.

## 5. Conclusions

In this paper, an underwater laser 3D scanning system based on rotation scanning is proposed, which is composed of line laser, underwater camera and turntable, realizing rotation scanning in the range of 50 ° (vertical) × 360 ° (horizontal). In order to achieve high-precision and high-resolution reconstruction results, this paper proposes a refraction correction algorithm for watertight devices, carries out three-dimensional reconstruction experiments on underwater targets, and obtains three-dimensional point clouds. Finally, in order to verify the feasibility of the system and the effectiveness of the algorithm, we use a standard ball with a radius of 20.01 mm for error analysis, showing that the underwater 3D reconstruction error of the system is less than 0.6 mm, which further proves that the line laser scanning system with low cost and simple structure can replace the expensive professional optical depth sensor and provide a new reference for underwater 3D laser reconstruction technology.

## Figures and Tables

**Figure 1 sensors-21-01402-f001:**
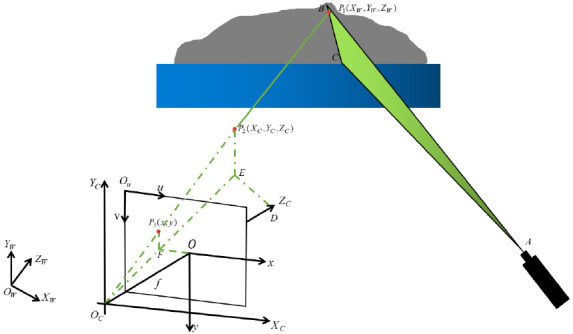
Geometric relations in coordinate system transformation.

**Figure 2 sensors-21-01402-f002:**
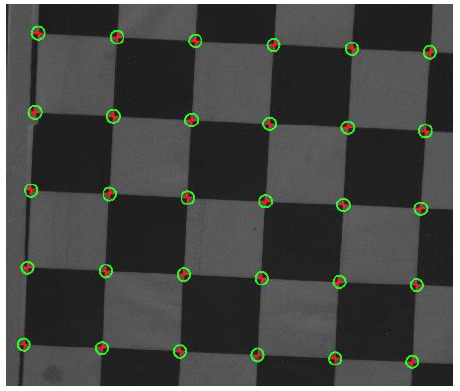
Some corners extracted by Harris algorithm.

**Figure 3 sensors-21-01402-f003:**
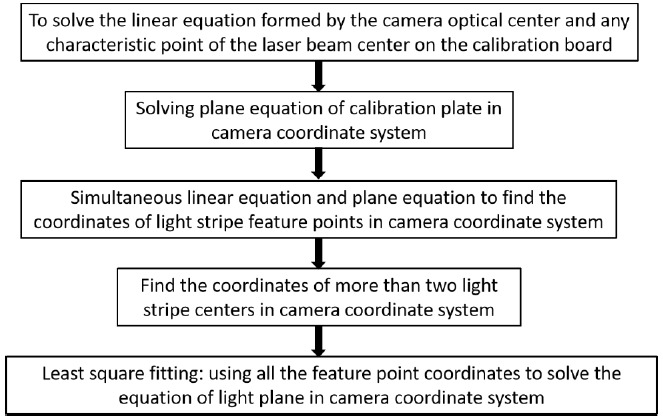
Flow chart of laser plane equation calibration.

**Figure 4 sensors-21-01402-f004:**
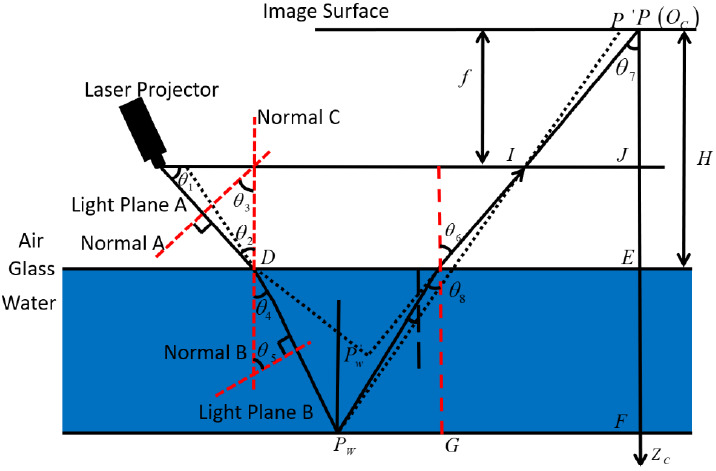
Refraction caused by watertight devices.

**Figure 5 sensors-21-01402-f005:**
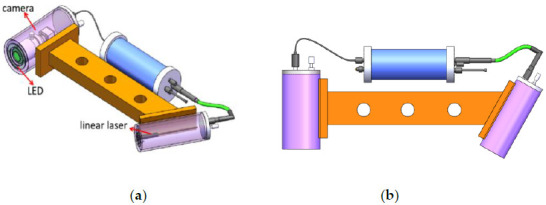
Mechanical model of the rotating scanning system. (**a**) Optical system; (**b**) System mechanical structure with variable included angle; (**c**) Rotating stage; (**d**) The overall model of the system.

**Figure 6 sensors-21-01402-f006:**
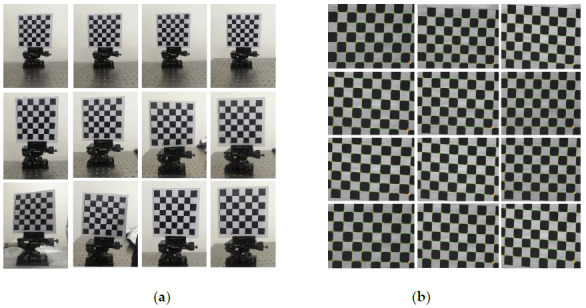
Camera calibration process in the air and some corner points extraction results. (**a**) Twelve Calibration boards with different attitude; (**b**) Corner extraction results of image.

**Figure 7 sensors-21-01402-f007:**
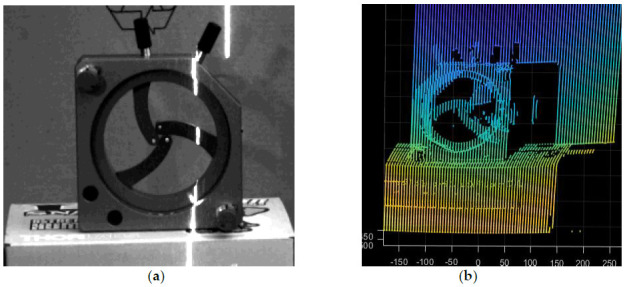
Three-dimensional (3D) reconstruction point cloud in air. (**a**) Measured object in the air; (**b**) Point cloud of object.

**Figure 8 sensors-21-01402-f008:**
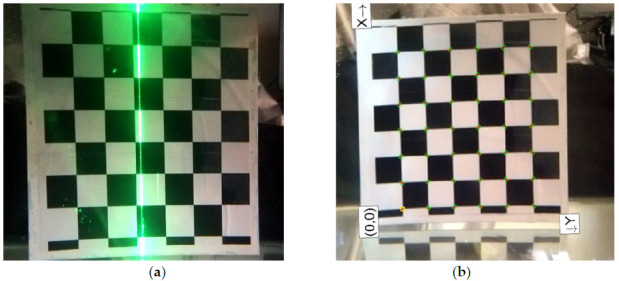
Underwater chessboard calibration. (**a**) Checkerboard calibration; (**b**) Corner extraction of underwater chessboard.

**Figure 9 sensors-21-01402-f009:**
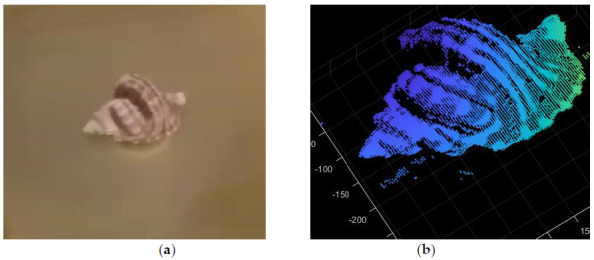
3D reconstruction results of underwater conch. (**a**) Conch placed in the water; (**b**) 3D point cloud of conch.

**Figure 10 sensors-21-01402-f010:**
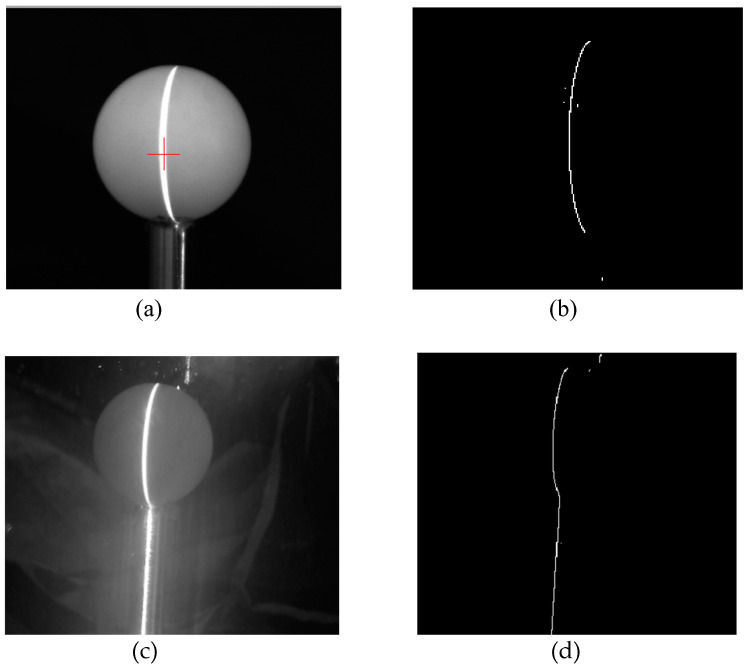
Measurement value and error of standard ball at different distances. (**a**) Reconstruction of standard sphere in air; (**b**) Extraction of laser stripe center in air; (**c**) Reconstruction of standard sphere in water; (**d**) Stripe center extraction of underwater standard sphere.

**Figure 11 sensors-21-01402-f011:**
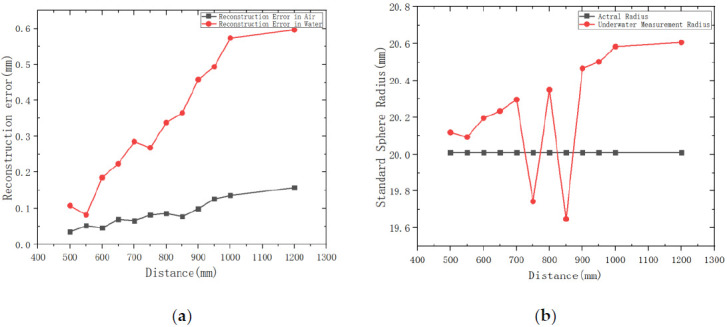
Relationship between the measured value of standard error sphere radius and the measured distance. (**a**) Reconstruction error results in water and air; (**b**) Reconstruction of standard ball diameter at different distances.

**Table 1 sensors-21-01402-t001:** Pixel coordinates of some corners of calibration board.

Index	u(pix.)	v(pix.)
1	1148.465576	1012.713867
2	800.38644746	1893.737305
3	497.802887	1882.058228
4	193.6052246	1868.337646
5	497.802887	1882.058228
6	264.0452271	363.3004456

**Table 2 sensors-21-01402-t002:** Mechanical model of rotary scanning system.

**Performance**	
Measuring Method	Triangulation
Scan Range	Minimum: 0.14 m | Maximum: 2.5 m
Field of View	360°Pan 90°Tilt
Operating Temperature	-10℃–40℃
Pan and Roll Accuracy	0.1°
Pitch and Roll Accuracy	±0.25°
**Laser**	
Wavelength	520 nm
Electric Current	≤300 mA
Power Supply	50 mW
**Camera**	
Model	WP-UT530M
Pixel Size	4.8×4.8 μm
Resolution	2592×2048
Frame Rate	60 FPS
Exposure Time	16 μs–1 s
Spectral Range	380–650 nm

**Table 3 sensors-21-01402-t003:** Some parameters of the system.

Parameter	Result (Pix.)
Focal Length	[$1.2329\times 10^{4}$ $1.229\times 10^{4}$]
Principal Point	[$1.3135\times 10^{3}$ $7.4849\times 10^{2}$]
Image Size	[2048 2592]
Radial Distortion	[−0.1380 15.7017]
Tangential Distortion	[0 0]

**Table 4 sensors-21-01402-t004:** Measurement result and error of standard ball.

Distance(mm)	R(mm)	Error(mm)	$R_{C}(\mathrm{mm})$	$E{\mathrm{rror}}_{c}(\mathrm{mm})$
500	24.541	4.531	20.117	0.107
550	24.995	4.985	20.091	0.081
600	25.336	5.326	20.195	0.185
650	25.365	5.355	20.233	0.223
700	24.379	4.369	20.295	0.285
750	25.52	5.51	19.742	0.268
800	25.993	5.983	20.347	0.337
850	26.238	6.228	19.646	0.364
900	26.862	6.852	20.468	0.458
950	25.999	5.989	20.503	0.493
1000	27.109	7.099	20.583	0.573
1200	27.022	7.012	19.414	0.596
Average	25.780	5.770	20.136	0.331
Max	27.109	7.099	20.583	0.596

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
