# Peer review of "Underwater High-Precision 3D Reconstruction System Based on Rotating Scanning"

_sensors, 2021, doi:10.3390/s21041402_

Round 1
Reviewer 1 Report
Abstract:
Need to specify that it is a range of field of view.
Not "surface of different media", but "interface between..."
1. "...triangle principle..." change to "...triangulation..."
"In addition..." change to "Also, ..."
"...mechanical method is..." change to "...was..."
"...methos is developed..." change to "...was..."
"...in some occasions..." change to "...on..."
"...bleier..." should start with a capital letter
"...puts forward..." change to "...proposes..." or "...describes..."
No need to explain what are intrinsic and extrinsic parameters - this is common knowledge.
Introduction of the term "internal/external parameters" is not necessary - it is the same as
intrinsics and extrinsics.
Harris algorithm for extraction of points with high cornerness is known to be significantly
inferior to other algorithms that allow for determination of corners' location with sub-pixel accuracy.
The authors have to explain why this technique was adopted.
2.4:
"...because of the required..." change to "...because of the requirement..."
Phrase "The influence of refraction usually includes two aspects" and the next one appear twice.
"...refraction correction of waterproof" what? Device? System?
The proposed refractive correction does not sound correct. It is known that camera calibration
in the presence of two flat refractive interfaces requires determination of three additional
parameters - for example, distance between focal point and nearest refractive interface and
normal to interface in the camera system (assuming that separation between interfaces is known).
One may assume that the same refers to the laser too if it is inside the housing. Raytracing through the image pixel can
be done directly, but the inverse process - determination of the pixel that records known point in water -
can be performed only iteratively.
Refraction may introdice insignificant distortions in certain cases, but neglecting its effects may lead to serious errors.
"Through Zhang Zhengyou's calibration method..." change to "Using..."
"...we can know..." change to "...we can determine..."
"...system can well complete..." change to "...system can complete..."
Line 255: "calibrate"
"It's easy to know..." change to "It is known..."
Light is not refracting AFTER passing media, it is refracting ON interfaces separating media with different refractive indices.
Underwater calibration does not make much sense. Better is to calibrate camera in-air and correct imagery for refractive distortions.
Author Response
-------------Reviewer Comments-------------
Abstract:
Need to specify that it is a range of field of view.
Not "surface of different media", but "interface between..."
1. "...triangle principle..." change to "...triangulation..."
"In addition..." change to "Also, ..."
"...mechanical method is..." change to "...was..."
"...methods is developed..." change to "...was..."
"...in some occasions..." change to "...on..."
"...bleier..." should start with a capital letter
"...puts forward..." change to "...proposes..." or "...describes..."
No need to explain what are intrinsic and extrinsic parameters - this is common knowledge. Introduction of the term "internal/external parameters" is not necessary - it is the same as intrinsics and extrinsics.
Harris algorithm for extraction of points with high cornerness is known to be significantly inferior to other algorithms that allow for determination of corners' location with sub-pixel accuracy.
The authors have to explain why this technique was adopted.
2.4:
"...because of the required..." change to "...because of the requirement..."
Phrase "The influence of refraction usually includes two aspects" and the next one appear twice.
"...refraction correction of waterproof" what? Device? System?
The proposed refractive correction does not sound correct. It is known that camera calibration
in the presence of two flat refractive interfaces requires determination of three additional
parameters - for example, distance between focal point and nearest refractive interface and
normal to interface in the camera system (assuming that separation between interfaces is known).
One may assume that the same refers to the laser too if it is inside the housing. Raytracing through the image pixel can
be done directly, but the inverse process - determination of the pixel that records known point in water -
can be performed only iteratively.
Refraction may introdice insignificant distortions in certain cases, but neglecting its effects may lead to serious errors.
"Through Zhang Zhengyou's calibration method..." change to "Using..."
"...we can know..." change to "...we can determine..."
"...system can well complete..." change to "...system can complete..."
Line 255: "calibrate"
"It's easy to know..." change to "It is known..."
Light is not refracting AFTER passing media, it is refracting ON interfaces separating media with different refractive indices.
Underwater calibration does not make much sense. Better is to calibrate camera in-air and correct imagery for refractive distortions.
First of all, thank you very much for taking the time to read and revise my article during your busy schedule. Thank you for your valuable comments, which will play a very important role in improving the quality of my paper. I have carefully studied the comments of the reviewers, and carefully revised the manuscript one by one according to the suggestions. The specific contents are as follows:
- Not "surface of different media", but "interface between..."
"...triangle principle..." change to "...triangulation..."
"In addition..." change to "Also, ..."
"...mechanical method is..." change to "...was..."
"...methods is developed..." change to "...was..."
"...in some occasions..." change to "...on..."
"...bleier..." should start with a capital letter
"...puts forward..." change to "...proposes..." or "...describes...""...because of the required..." change to "...because of the requirement...""Through Zhang Zhengyou's calibration method..." change to "Using..."
"...we can know..." change to "...we can determine..."
"...system can well complete..." change to "...system can complete..."
Line 255: "calibrate"
"It's easy to know..." change to "It is known..."
Light is not refracting AFTER passing media, it is refracting ON interfaces separating media with different refractive indices.Phrase "The influence of refraction usually includes two aspects" and the next one appear twice.
Reply:
Please excuse this negligence in the manuscript and the inadequacy of the self-review of the manuscript. I am very ashamed for the inconvenience and trouble caused to you in the review. We have made serious corrections to the existing errors in word and tense of sentences in the manuscript.
- Abstract:
Need to specify that it is a range of field of view.
Reply:
The range of field of view of the system is 50 ° (vertical) × 360 ° (horizontal), and we have specified it in the manuscript.’… The underwater 3D data acquisition can be realized in the range of field of view of 50 ° (vertical) × 360 ° (horizontal).…’.
-No need to explain what are intrinsic and extrinsic parameters - this is common knowledge. Introduction of the term "internal/external parameters" is not necessary - it is the same as intrinsics and extrinsics.
Reply:
Thank you for your suggestion. We will delete unnecessary noun explanations in the manuscript to make the article more concise
-Harris algorithm for extraction of points with high cornerness is known to be significantly inferior to other algorithms that allow for determination of corners' location with sub-pixel accuracy.
The authors have to explain why this technique was adopted.
Reply:
At present, Harris, SIFT, FAST and other corner extraction algorithms are widely used in 3D reconstruction. Among them, SIFT and FAST are the optimization algorithms based on Harris, which are rich in information and suitable for matching in massive feature database. They have high speed, especially the optimized SIFT can basically achieve the real-time extraction of large objects, and can also generate a large number of feature data for a few flat objects. The advantage of FAST feature detection is much faster than other algorithms in speed, but the disadvantage is that it is greatly affected by image noise and the set threshold; the corner detected is not necessarily optimal, mainly because it is affected by the corner distribution; the feature detected by fast feature does not have direction information, and the fast feature does not have direction information Feature points do not produce multi-scale features, so they do not have rotation invariance. In this paper, the corner detection algorithm is used to extract the pixel coordinates of the corners on the chessboard in the process of camera calibration, to achieve the calculation of camera parameters, so as to obtain the conversion relationship from the two-dimensional coordinate system to the three-dimensional coordinate system. There is no need for high-speed extraction and large amount of data, and in underwater work, due to the instability of the working environment, a large number of feature data can not guarantee its significance. Although the Harris algorithm has a general recognition effect for complex multi-edge scenes, it can well recognize the grayscale changes and translational rotation changes of flat images, and it also has the invariance of illumination and partial invariance of ratio changes. Because Harris corner detection operator uses the gray second-order moment matrix of the area near the corner. The second-order moment matrix can be expressed as an ellipse, and the long and short axis of the ellipse are exactly the inverse of the square root of the eigenvalue of the second-order moment matrix. When the characteristic ellipse rotates, the characteristic value does not change, so the response value R of the judgment corner does not change, which shows that the Harris corner detection operator has rotation invariance.When performing Harris corner detection, we use a differential operator to differentiate the image, and the differential operation is not sensitive to the increase or contraction of the image density and the increase or decrease of the brightness. In other words, the affine transformation of brightness and contrast does not change the position where the extreme point of Harris response appears.This paper focuses on the reconstruction system and refraction correction in the reconstruction process, and does not verify whether various high-precision corner detection algorithms are suitable for underwater environment. Thank you for your suggestions. We will continue to improve the algorithm in the follow-up work, and compare the effect of Harris corner detection algorithm and other algorithms in underwater environment.
-"...refraction correction of waterproof" what? Device? System?
Reply:
In the manuscript, we have modified the refraction correction of the waterproof to the refraction compensation of the 3D reconstruction system
-The proposed refractive correction does not sound correct. It is known that camera calibration in the presence of two flat refractive interfaces requires determination of three additional parameters - for example, distance between focal point and nearest refractive interface and normal to interface in the camera system (assuming that separation between interfaces is known). One may assume that the same refers to the laser too if it is inside the housing. Raytracing through the image pixel can be done directly, but the inverse process - determination of the pixel that records known point in water -can be performed only iteratively. Refraction may introduce insignificant distortions in certain cases, but neglecting its effects may lead to serious errors.
Underwater calibration does not make much sense. Better is to calibrate camera in-air and correct imagery for refractive distortions.
Reply:
Thank you very much for your query. After careful consideration and literature review, we will give the following answers and attach corresponding references.
In the experiment, the whole system is watertight and placed underwater to reconstruct the underwater target and terrain. In the process of work, the system will face many problems such as refraction, scattering and attenuation of water, among which refraction is one of the key problems. In underwater applications, the camera is usually sealed in a waterproof cover with a glass window. When shooting, the light reflected by the object will enter the camera lens through three media: water, glass waterproof cover and air. Due to the refraction of light on the contact surface of different media, the laser plane will deflect on the contact surface, and the imaging points on the CCD target surface of the camera can not well represent the real position of the measured object, so the mathematical model on land is no longer established, and the calibration method is no longer suitable. The selected references are as follows:
- Yau T, Gong M L , Yang Y H. Underwater camera calibration using wavelength triangulation [C]// 2013 IEEE Conference on Computer Vision and Pattern Recognition, June 23-28, 2013: 2499-2506.
- Agrawal A, Ramalingam S, Taguchi Y, et al. A theory of multi-layer flat refractive geometry [C]// 2012 IEEE Conference on Computer Vision and Pattern Recognition, June 16-21,2012, Providence, RI. New York: IEEE, 2012: 3346-3353.
- Treibitz T, Schechner Y, Kunz C, et al. Flat Refractive Geometry[J]. IEEE Trans Pattern Anal Mach Intell, 2012, 34(1):51-65.
- Xie Zexiao, Yu Jiangshu, Chi Shukai, Li Junpeng, Li Meihui. Underwater calibration and measurement of nonparallel binocular vision system [J]. Acta optica Sinica, 2019,39 (09): 195-204.
We have modified the section 2.4 of the article. In order to better express our ideas, we divide the waterproof device correction method into the refraction compensation of the laser plane equation and the CCD image plane coordinates. The plane equation and two-dimensional pixel coordinates after refraction are calculated respectively, and then substituting them into the land-based 3D measurement model, the camera calibration parameters after refraction compensation can be obtained, and then more accurate 3D coordinates can finally be calculated.

Reviewer 2 Report
In this manuscript, the authors investigated an underwater high-precision line laser three-dimensional scanning system with rotational scanning composed of a low illumination underwater camera and a green line laser projector. To reduce the viewing angle error caused by the refraction of light on the surface of different media and reduce the impact of refraction on the image quality, the authors implemented a refraction correction on the waterproof device of the system.
This issue is relevant, considering that underwater high-precision 3D reconstruction can be potentially applied is several fields, such as underwater military rescue work, in marine scientific research and so on.
However, in my opinion, the Introduction section is confused and the motivation for this work is not well emphasized. One major issue I do have with the manuscript is the quality of English. While I don’t believe it makes the work unreadable there are many errors littered through the manuscript and in a couple of cases it can obscure the authors point.
Moreover, there are several other points that should be fixed:
- The sentence on lines 27-34 is too long and confused. I suggest to clarify and better explain these concepts, especially considering that they are part of the motivations of this work.
- On line 46: What does it mean "and so on"? If the authors want to describe a typical setup, they cannot just put two elements (i.e., the laser and the camera) and then write "so on".
- On line 48: It is not clear how the plane equation of light plane in camera coordinate system can be obtained. I suggest to insert some references or to better explain this.
- On line 57: Usually, reference is cited as: Bleier et al. Please correct all the citations in the text.
- On line 59: The reference corresponding to Jules S. Jaffe is missing.
- On line 85: The sentence is incomplete!
- In Figure 1 O is reported as Ou, please correct.
- In Eq. (1): please specify what are R and T.
- On line 119: “this article” is repeated twice.
- On line 136: Regarding the Harris corner detection algorithm, can the authors add a reference to this algorithm and briefly describe the basic working principle? It could be useful for readers not expert in this field but interested to the results presented in this study.
- On line 145: Please, insert citations of the calibration methods reported here.
- On lines 152-153: The sentence is incomplete!
- Subsection 2.4: Really confused. Please organize better the section to improve the readability.
- On line 175 specify: the normal to the glass layer.
- On line 175, regarding θ3, maybe the authors would like to say "is the angle between the normal to A and the normal to C"
- Eq. (5): n1 and n3 are the refractive index of what?
- On lines 233-235: This sentence is not clear.
- In the caption of Figure 12 part of the text is repeated twice.
- On line 315: “…standard sphere radius is 0.596”, please insert the unit (i.e., mm).
- On line 332 “ion technology” is repeated.
- Sometimes the template requirements are not met, please check carefully all the text.
Considering all these reasons, I recommend this article for publication after major revisions.
Round 2
Reviewer 1 Report
Authors' comments:
SIFT is not based on Harris detector: the former uses DoG, the latter second-moment matrix. Also, SIFT is scale- and intensity-invariant, so Harris detector has no advantages in this sense.
References for image distortion due to refractive effects have been added, but it is still not clear how the authors achieve refractive "correction" without specialized calibration targeting refraction-related parameters and iterative
solution for correction value. The technique probably should be called "reduction of refractive distortions", not "correction".
Line 161: Equation cannot be formed by "...the center and ...the point" - it is a mathematical abstraction. Are the authors talking about a line or a plane that can be described by the equation?
Line 163: "...plane equation..." does not sound right. Maybe "equation that describes the plane"?
Line 171: "imaging system" is not necessarily a photographic camera. It could be a sonar or multispectral camera, for instance.
Line 173: "is finally BEING imaged"
Line 176: "intersection point" of what? Also, "intersection point" cannot be "the true three-dimensional information". It can only carry information.
Line 180: the authors assume that the refractive surfaces are parallel to the focal plane (sensor). This assumption greatly simplifies the calculations but rarely can be achieved in practice. Was this assumption verified experimentally?
Reviewer 2 Report
In my view, the authors, within the scope of this manuscript, have sufficiently addressed the concerns I raised in my review in their revised version. Therefore, I believe that the revised manuscript is acceptable for publication.
Author Response
Thank you for your comments concerning our manuscript,those comments are all valuable and very helpful for improving our paper.